# Study on the Mechanism of a Hanging Roof at a Difficult Caving End in a Fully-Mechanized Top Coal Caving Face

Hongtao Liu [1],*, Cheng Hao [1], Zijun Han [1], Qinyu Liu [1], Haozhu Wang [1], Jialu Liang [1] and Dandan Zhang [2]

1   School of Energy and Mining Engineering, China University of Mining and Technology, Beijing 100083, China
2   School of Emergency Management and Safety Engineering, China University of Mining and Technology, Beijing 100083, China
*   Correspondence: 108925@cumtb.edu.cn

**Abstract:** During the mining process of fully-mechanized caving faces, the roof of the roadway behind the working face easily forms an arched-shape hanging roof structure with the working face pushed forward, which results in potential hazards such as gas accumulation and large-scale roof collapse. Based on the actual situation of a hanging roof at a difficult caving end in fully-mechanized top coal caving faces, through borehole exploration, surrounding rock displacement observation, bolt stress monitoring, theoretical formula calculation, and numerical simulation methods, the structure characteristics of the hanging roof at the end of the fully-mechanized caving face are studied. The ultimate failure depth and ultimate break distance of the hanging roof structure at the end of the working face are obtained, and its formation mechanism is analyzed. It is concluded that the hanging structure is formed by the following reasons: the lithology of sandy mudstone and fine sandstone above the top coal of the roadway is strong; the hanging roof structure is less affected by working-face mining; there is a result of insufficient rotary pressure of the upper mudstone while working together with the protective coal pillar and end support the caving step distance of the curved hanging roof structure is 10~13.55 m.

**Keywords:** end hanging roof; caving mining method; roof caving; numerical simulation





## 1. Introduction

In coal mining, with the working face forward, the periodic fracture of the roof will form an arc-shaped hanging roof at the end of the working face [1,2]. The existence of a certain range of arc triangular plate structure is conducive to the stability of the roadway end and ensures the safety of the mining face. However, when the hanging roof area is too large, there will be potential safety hazards such as gas accumulation and large-scale roof collapse. The arched-shape hanging roof easily causes high underground stress and the accumulation of gas and dust [3,4]. When the hanging roof structure rotates and breaks, a large area of pressure will be generated, leading to rock burst, coal and gas outburst, or large deformations in roadways. On the other hand, in the process of moving, the end support will inevitably rub and collide with the end of the bolt at the top of the roadway, which may produce sparks and lead to the explosion of gas accumulated at the end, causing unpredictable disasters [5–11].

As one of the common problems in mine disasters, the end-hanging problem has been studied by many scholars.

Wang et al. [12] took the hard roof as the research object and, using a mechanical model and theoretical research methods, obtained a reasonable length of the hanging roof of the working face end roadway under the premise of safe production.

He et al. [13] selected the key strata in the roof of a mined-out area as the research object and simplified it into the rock beam model for analysis. Considering the bending deformation in roof strata, the creep model is used to discuss the characteristics of buckling load time and instability conditions.

Wang et al. [14] discussed the rheological properties of materials under different conditions and obtained three cases of delayed instability of hanging rooves in goaf.

Wang et al. [15] analyzed the structure of the curved triangular plate of the roof after mining, established the mechanical model of the rectangular plate structure, determined the failure position of the hanging plate, and explained the generation mechanism of the curved triangular plate.

Qin et al. [16], based on the thin plate theory of elasticity, analyzed the rectangular plate structure with fixed support on both sides. It is considered that the hanging roof structure can be simplified as a thin plate with two sides fixed. The bending moment calculation formula of the hanging plate structure is obtained by using the generalized simply-supported edge concept and the superposition algorithm.

Luan et al. [17] established the mechanical model of gob-side entry retaining, analyzed the relationship between the characteristics of mine pressure behavior and the arc triangle plate at the end of the retaining roadway, and proposed the position prediction and measurement method of the arc triangle plate.

Liu et al. [18,19] analyzed the basic roof structure of the working face end, used a three-parameter Weibull function to analyze the two-way abutment pressure in the triangular area of the end, and put forward the transfer method of two-way abutment pressure.

Xu [20] studied the coupling relationship between hydraulic support and roof strata by means of theoretical analysis and numerical simulation and analyzed the hanging mechanism of the arc triangular plate at the end of the working face in the Jinjitan coal mine.

Pavlova et al. [21] used a numerical simulation research method to the influence of different length of the hanging roof, and the relationship between the displacement and stress distribution of the working face and the length of the hanging roof is obtained, which provides a basis for formulating preventive measures for the collapse of the hanging roof structure.

Xue et al. [22] used the research method of numerical simulation to discuss the influence of the length of the hanging roof and periodic weighting step on the stress in front of the coal wall, the classification prediction method of rock burst hazard is established, and the countermeasures for different hazard levels of rock burst are proposed.

Dychkovskyi et al. [23] used SolidWorks 2019 software to evaluate stress of a mine field in terms of Lvivvuhillia SE mine and analyzed the properties of the support pressure zone formed in front of the stope and along the length of the pillar. It is considered that, if the stope is in the same plane, its front support pressure area and the wall of the development work area will connect to each other. Based on the analysis of the stress and strain state of rock mass in the process of double unit longwall mining, the effective mining parameters and the control method of rock pressure are determined.

By analyzing and evaluating the geological conditions, technological parameters, and support methods of longwall mining and using methods of statistics, investigation, and analysis of field data, Vu [24] pointed out the causes and laws of spalling and roof fall in longwall fully-mechanized mining faces. In addition, some measures to prevent spalling and roof fall are put forward.

Chen et al. [25], based on the research background of thick and hard hanging roof in Datong mining area, established a mechanical model of roof periodic collapse based on load and instability characteristics, and the relationship between the breaking distance, breaking angle, and the working resistance of the support is studied.

In the previous studies, most of them studied the reasonable length of the hanging roof structure and its impact on the stress of the roadway, or their theoretical calculations, but few of them studied the mechanism of the hanging roof structure from multiple angles, which combines field practice, theoretical analysis, and numerical simulation. The purpose of this paper is to study the collapse form and collapse mechanism of the end arched-shape hanging roof structure. The research results can provide reference for the roof control of the working face end in the process of advancing the working face, so as to ensure the safety of the working face end and ensure the safe production of the working face.

## 2. Background

### 2.1. Mining Conditions of the Working Face

As shown in Figure 1, Wangjialing coal mine is located in the Shanxi Province, its area is 119.71 km², including 5 layers of recoverable and locally recoverable coal seams. The main mining coal seams are No. 2 coal seams and No. 10 coal seams, which can be mined in the whole area. The dip angle of coal seams is small and does not change much, with an average of 3°~5°. The roof of the coal seam is stable, the geological structure is relatively simple, the hydrogeological conditions are simple, the water inflow is small, the gas content is low, and the mining technical conditions are good.

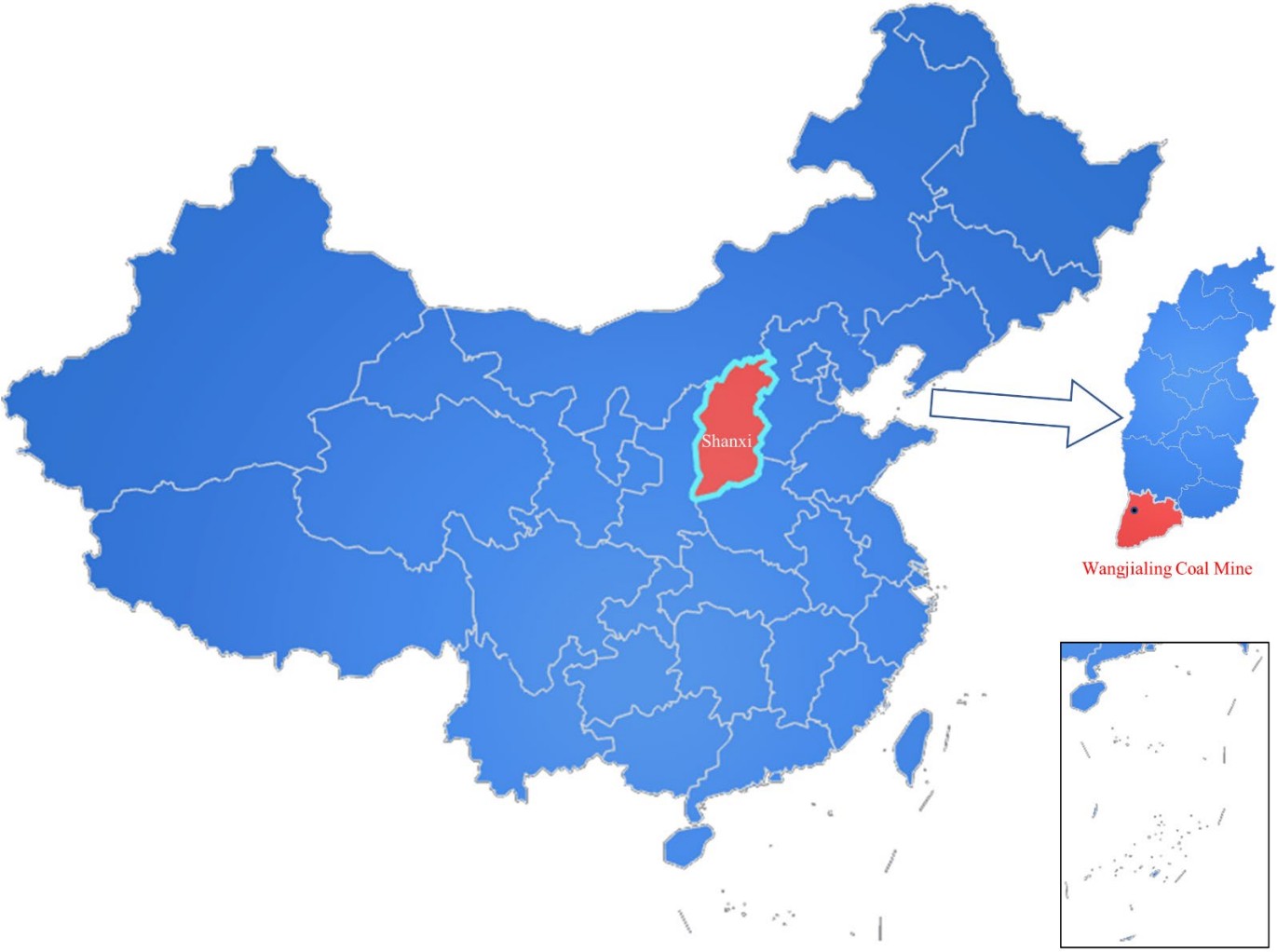

**Figure 1.** The location of Wangjialing coal mine.

At present, the mine is mining the No. 2 coal seam, with a thickness of 3.09–8.50 m at an average of 6.20 m. The coal seam generally contains 1–2 layers of dirt band. The roof of the coal seam is sand mudstone, above is fine sandstone, the floor is sand mudstone, the comprehensive column is shown in Figure 2a, and the local floor contains fine sandstone and quartz sandstone.

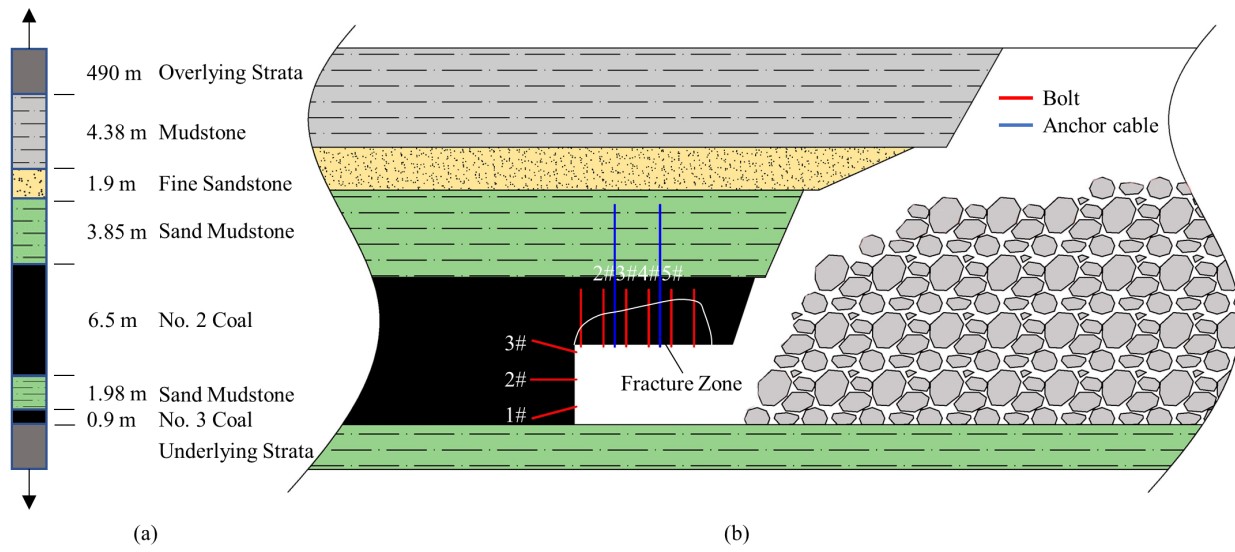

**Figure 2.** (**a**) Comprehensive column of the roof and floor of the No. 2 coal seam; (**b**) the schematic diagram of the hanging roof at the end of the goaf.

The Platts coefficient of the No. 2 coal seam is about 1.5, the coal seam is soft, the joint fissure is developed, and the top coal caving is good. In the process of working-face mining, a large area of hanging roof appears in the end area of the working face, as shown in Figure 2b, which brings great safety hazard to mine production.

The plan of the working face is shown in Figure 3. The study working face is located in the northern area of its panel. It is arranged in two lanes, which are composed of a haulage roadway, return air roadway, cut hole, and related roundabout and chamber. The average buried depth of the working face is 500 m, the length is 299.6 m, and the advancing length is 3316 m. The working face adopts fully-mechanized top coal caving mining technology. The mining height is 3.1 m, the average height of coal caving is 3.23 m, the mining ratio is 1:1.04, and the cycle progress is 0.85 m. Both sides of the working face are not mined.

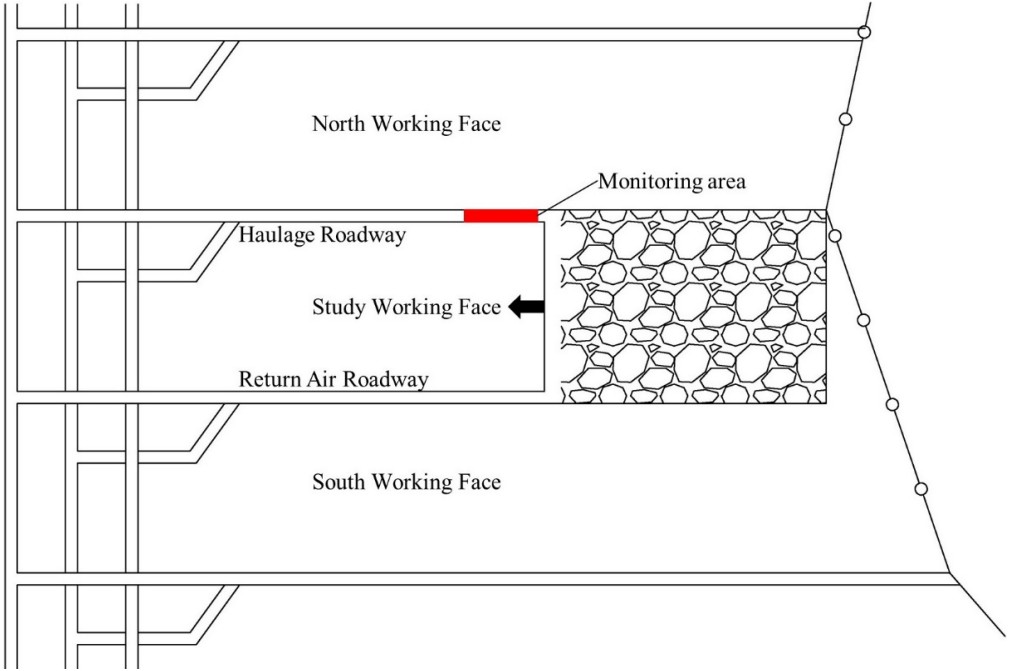

**Figure 3.** The plan of the working face layout.

### 2.2. Roadway Support Parameters

The working face's haulage roadway has a width of 5.6 m, height of 3.1 m, and uses "anchor-net-cable" support; roadway roof-specific support measures are as follows:

Top bolt: φ20 × 2500 mm left-handed thread steel is used, and the row spacing is 1000 × 900 mm. In addition, φ14 mm steel beam is used to connect the top bolt at the same time in the roadway roof, and the top mesh is φ6 mm steel mesh. The bearing capacity of the bolt is 10 tons.

Top anchor cable: φ17.8 × 6250 mm steel strand, spacing of 2000 × 1800 mm, 2-1-2 arrangement. The bearing capacity of the anchor cable is 18.6 tons.

It can be found from the support conditions of the haulage roadway in the working face that the support strength of the roadway roof is large, and there are 1.5 anchor cables and 6 bolts in the roof of the roadway per meter on average. According to the structural characteristics of the roadway roof strata mentioned above, the supporting strength of the haulage roadway in the working face is too large, and the roadway is only affected by one mining. The influence of mining on the roadway roof is limited, so it is easy to form a hanging structure at the end of the working face during the advance of the working face.

### 3. Study on the Law of Ground Pressure Behavior in the Working Face

#### 3.1. Deformation Characteristics of Surrounding Rock in a Mining Roadway

In order to observe the deformation degree of the surrounding rock of the roadway during the mining process of the working face, the mine pressure monitoring station is arranged at 40 m ahead of the working face of the haulage roadway in the working face, the displacement of the two sides of the roadway and the roof and floor during the advancement of the working face are continuously observed, and the observation results are recorded. The layout of the observation points is shown in Figure 3, and the observation results are shown in Figure 4.

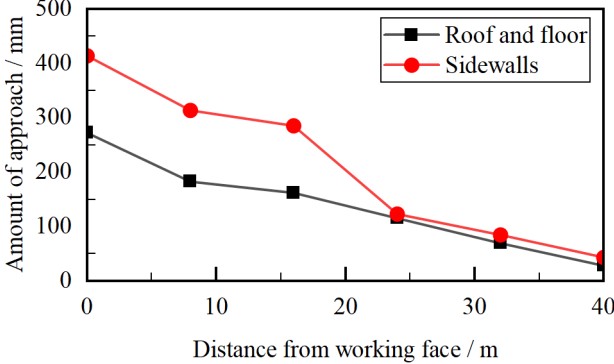

**Figure 4.** Surrounding rock deformation curve of the haulage roadway.

From Figure 4, it can be concluded that, during the observation period, as the working face advances forward, the displacement of the two sides of the roadway and the roof and floor is increasing, and the rate is growing. When the working face advances to the 5 m measuring point, the displacement of the two sides of the roadway and the roof and floor is the largest, and the maximum displacements are 415 mm and 274 mm, respectively. The roadway is mainly characterized by the bulging of the two sides, and the deformation of the roof and floor is not obvious.

Combined with the actual roof hanging at the end of the working face, under the influence of working-face mining, the maximum subsidence of the roadway roof is 274 mm. At this time, the anchor cable is in a load-bearing state and plays a major role.

#### 3.2. Roof Failure Characteristics of the Mining Roadway

In order to find out the damage of the roadway roof in the mining process of the working face and the change of the overlying strata of the roadway roof, five monitoring

stations were arranged along the haulage roadway of the working face from 20 m away from the working face, with a spacing of 20 m. The roof peeper was used to study the roof peep, and the peeping results are shown in Figure 5.

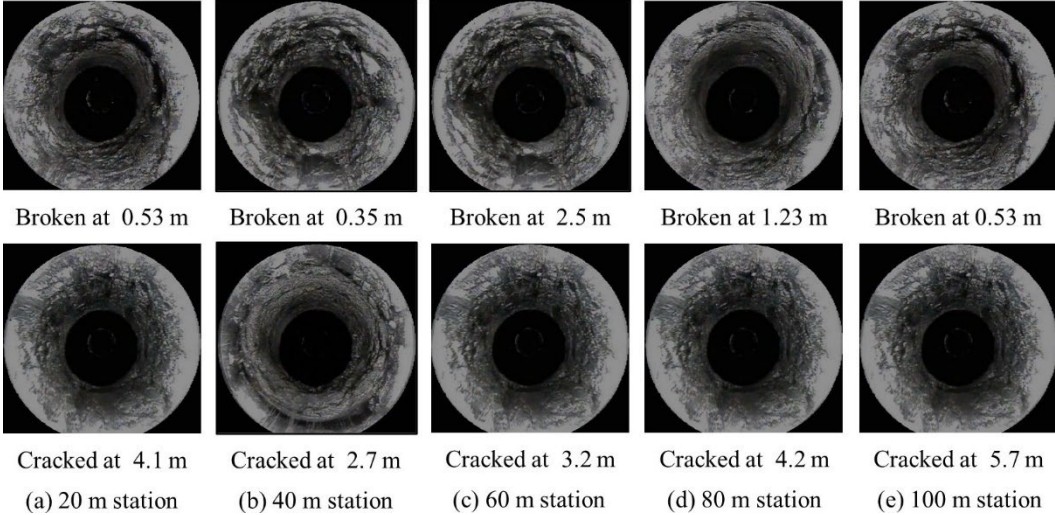

Broken at 0.53 m    Broken at 0.35 m    Broken at 2.5 m    Broken at 1.23 m    Broken at 0.53 m

Cracked at 4.1 m    Cracked at 2.7 m    Cracked at 3.2 m    Cracked at 4.2 m    Cracked at 5.7 m

(a) 20 m station    (b) 40 m station    (c) 60 m station    (d) 80 m station    (e) 100 m station

**Figure 5.** Detecting breaks and cracks at several monitoring stations placed apart from the working face and spaced by 20 m.

From Figure 5, it can be found that, when the distance from the working face is about 20 m, the shallow roof of the roadway is broken at 0.53 m, slight cracks appear at 4.1 m, and the deep rock layer of the roof is relatively complete. When the distance from the working face is 40 m, the roadway roof is broken near 0.35 m and cracks appear at 2.7 m of the roadway roof. From the 40 m station, the deep rock strata of the roadway roof are complete, and there is no obvious fragmentation. With the increase of the distance from the working face, the integrity of the roadway roof is better, and no obvious broken zone and cracks are found in the roof. The shallow-layer damage at 40 m and 100 m away may be caused by the influence of mining on the primary fissure of rock mass. The mine pressure of the roadway is not obvious, and the surrounding rock is in a stable state.

The results of the peep are summarized, and the changes of the rock strata at each peep point are analyzed, as shown in Table 1. Based on this, the rock stratum structure diagram of the haulage roadway roof of the working face is drawn, as shown in Figure 6.

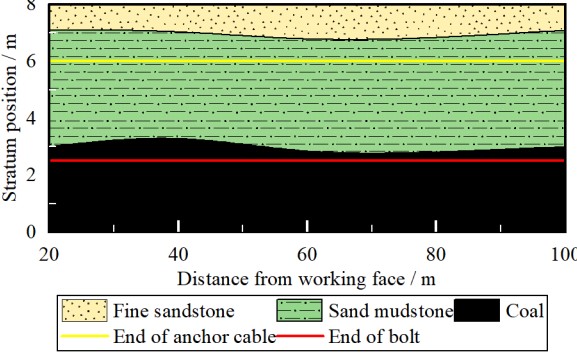

**Figure 6.** Schematic diagram of roof rock structure of roadway.

**Table 1.** Summary of roof peeping results.

| Rock Stratum | STA [1]-20 | | STA-40 | | STA-60 | | STA-80 | | STA-100 | |
| | D [2] /m | GT [3] /m | D /m | GT /m | D /m | GT /m | D /m | GT /m | D /m | GT /m |
|---|---|---|---|---|---|---|---|---|---|---|
| Fine sandstone | 0.9 | 8 | 0.9 | 8 | 1.3 | 8 | 1.2 | 8 | 0.9 | 8 |
| Sand mudstone | 4.1 | 7.1 | 3.6 | 7.1 | 4 | 6.7 | 4 | 6.8 | 4.1 | 7.1 |
| Coal | 3 | 3 | 3.5 | 3.5 | 2.7 | 2.7 | 2.8 | 2.8 | 3 | 3 |

[1] STA-station; five monitoring stations were arranged along the haulage roadway at 20 m, 40 m, 60 m, 80 m, and 100 m away from the working face. [2] D-depth; the depth of each stratum according to the results of roof peeping. [3] GT-grand total, the depth of the count of overlying strata according to the results of roof peeping.

From Figure 6, it can be seen that the roof rock structure of the roadway changes little, the rock layer is fixed, and there is no obvious layer change. Roof bolts are all in the top coal seam, the end of the local roof bolt is close to the coal–rock interface, and there is a risk of failure. The end of the anchor cable is all in the sand mudstone in the upper part of the top coal, and the end of the anchor cable is far from the boundary between the sand mudstone and the top fine sandstone, which is in a stable layer. Combined with the results of borehole peeping, the strata with broken zones and fracture are located in the shallow part of the roadway top coal or the junction of the roadway top coal and sand mudstone. Therefore, under this geological condition and support strength, the damage depth of the roadway roof is small and the integrity of the roadway roof is better after mining.

*3.3. Stress Characteristics of the Bolt in the Mining Roadway*

In the range of 5~40 m in the advanced face of the haulage roadway, a measuring point is arranged every 5 m, the measurement is carried out along the advancing direction of the working face, and the stress of the bolts at each position is recorded. The stress of the three bolts in the auxiliary side of the same roadway section and the four bolts in the middle of the roof are measured. In order to facilitate the statistical results, the bolts in the side and the roof are respectively numbered clockwise. The position and number of the measured bolt are shown in Figure 2. The measurement results are shown in Figure 7.

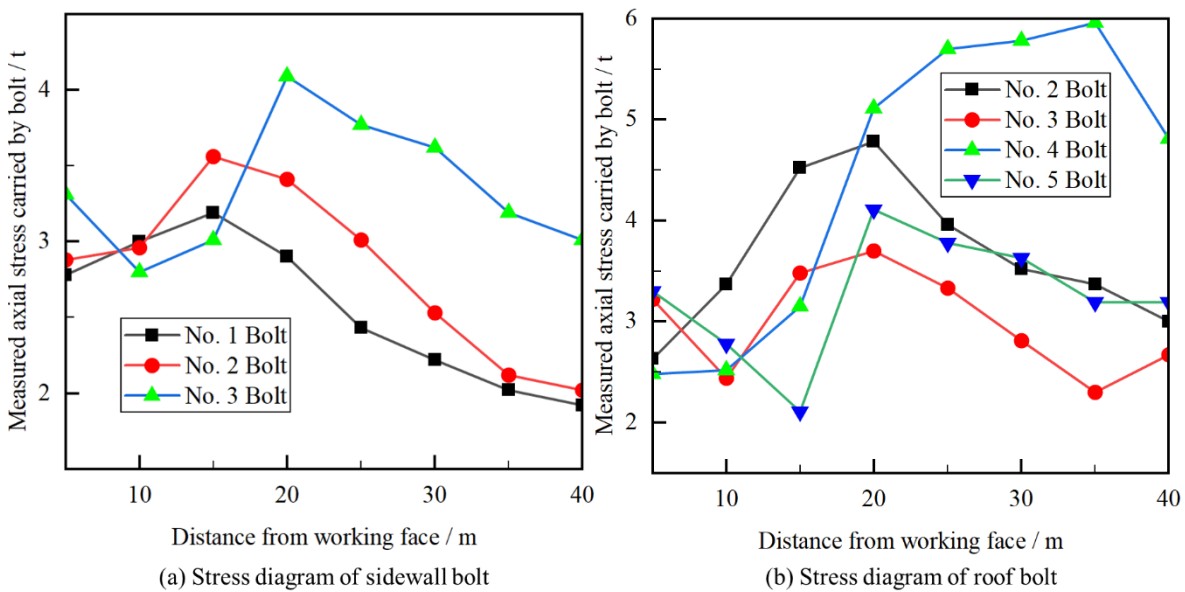

(a) Stress diagram of sidewall bolt     (b) Stress diagram of roof bolt

**Figure 7.** Stress diagram of bolts. (**a**) The stress diagram of a sidewall bolt; (**b**) The stress diagram of a roof bolt.

From Figure 7, it can be concluded that:

(1) It can be seen from the stress diagram of the sidewall bolt that the stress of the bolt in the top of the roadway side is larger than the other bolts, which reaches the maximum at 20 m from the working face, and the peak value is 4.1 t. Induced stress on the bolt falls as distance from the working face increases.

(2) According to the stress diagram of the roof bolt, the stress of the bolt (e.g., No. 4) in the middle of the roadway roof is greater than that on both sides (e.g., No. 2 and No. 5) of the roadway roof; see Figure 2. The stress reaches the maximum near 35 m from the mining face, and the peak value is 5.9 t. As the distance from the working face increases, the stress of some bolts (e.g., No. 2, No. 3, and No. 5) on the roof of the roadway tends to be stable at about 3 t.

(3) Generally speaking, in the monitoring area, the stress of the bolt at the top of the roadway is greater than that of the bolt at the side of the roadway, but the stress of the bolt at the roof and the side of the roadway is small, and there is a certain distance from the ultimate bearing capacity of the bolt. In addition, the working face roadway roof stability is good and less affected by working-face mining, with a high safety factor.

Combined with the end hanging roof structure, it can be found that the bolt in front of the working face is less stressed and in a stable state after being affected by the mining of the working face, far from reaching the breaking limit. After the mining of the working face, the roof of the roadway at the end of the working face has good integrity and stability under the suspension of the bolt and anchor cable. The roof of the roadway at a certain distance behind the working face is unstable and collapsed by the rotary pressure of the rock strata above the goaf.

### 3.4. Working Face End Hanging Structure Characteristics

Due to the limitation of field working conditions, the actual damage and stress of the end suspension structure cannot be directly measured, and only remote observation can be carried out. The end suspension structure mainly has the following characteristics:

(1) The hanging roof structure collapses periodically with the advance of the working face, and it collapses first at the end support edge and the corner of the coal pillar edge. The size of the caving range does not change much. The falling rock mainly comes from the top coal left by top coal caving and the sand mudstone above a small part of the top coal. The falling rock makes it difficult to fill the roadway behind the working face.

(2) In the hanging roof structure, the length of top coal is short, and most of the top coal without bolt support gradually collapses with the advance of the working face. The cantilever of the sand mudstone part above the top coal is longer. With the advancement of the working face, the sand mudstone cantilever will periodically collapse.

(3) With the advance of the working face, the shape of hanging roof structure is similar to arched-shape. In the axial direction of the roadway, the length of the arched-shape structure is about 10 m, and the arched-shape structure moves forward with the advance of the working face.

Combined with the previous measurement of the surrounding rock deformation of the haulage roadway in the working face, the peep of the roof rock structure, and the detection of the stress state of the roadway bolt, the analysis of the hanging roof characteristics of the working face end can be obtained:

Under the influence of working-face mining, the roadway roof in front of the working face has a maximum subsidence of 274 mm, and the maximum damage depth of the roof is about 2.7 m. It can be considered that, under this geological condition, after the roadway in front of the working face is affected by the mining of the working face, the integrity of the roof strata is good and there is no large-scale crushing and roof subsidence. After the working face is pushed over, it is easy to form a hanging roof structure near the end under the support of the end support and coal pillar.

## 4. Study on Working Face End Hanging Roof

*4.1. Theoretical Calculation of Failure Depth and Breaking Distance*

The length of the hanging roof is one of the key factors inducing roadway roof disasters [26,27]. It is generally believed that the supporting effect of bolt and anchor cable is mainly to suspend the broken rock strata in the roadway roof above the upper stable rock strata, as shown in Figure 2. In the supporting structure, the anchor cable plays a major role in suspension, so the condition for the collapse of the suspension structure is that the weight G of the rock stratum in the broken area is greater than the bearing capacity F of the anchor cable, that is, G > F.

The haulage roadway of the working face adopts the combined support of "anchor-net-cable". The anchor cable at the top of the roadway adopts the steel strand with the specification of $\varphi 17.8 \times 6250$ mm (the cross-sectional area is 193.45 mm$^2$ and the tensile strength is 1860 MPa), and the row spacing is $2000 \times 1800$ mm, 2-1-2 arrangement. According to the general calculation method, it can be considered that the number of anchor cables in 1 m roadway is 1.5, and the bearing capacity F provided by anchor cables in 1 m roadway is:

$$F = n \cdot P \cdot S \tag{1}$$

In the formula, n is the number of anchor cables in 1 m roadway, P is the tensile strength of anchor cables, and S is the cross-sectional area of anchor cables.

After calculation, the F value is $5.40 \times 10^5$ N.

The weight of broken coal-rock mass $G_{coal}$ and $G_{rock}$ in the plastic zone of 1 m roadway roof is:

$$G_{coal} = m_{coal} \cdot g = \rho_{coal} \cdot V_{coal} \cdot g \tag{2}$$

$$G_{rock} = m_{rock} \cdot g = \rho_{rock} \cdot V_{rock} \cdot g \tag{3}$$

In the formula, $m_{coal}$ and $m_{rock}$ are the weight of broken coal and rock mass, $\rho_{coal}$ and $\rho_{rock}$ are the density of broken coal and rock mass, $V_{coal}$ and $V_{rock}$ are the volume of broken coal and rock mass, and g is the gravity acceleration.

The height of the top coal is selected according to the average height of coal drawing, which is 3.23 m. The density of the coal body is 1390 kg/m$^3$, and the density of the rock mass is 2660 kg/m$^3$.

In addition, the $G_{coal}$ value is $2.46 \times 10^5$ N, the $G_{rock}$ value is $1.46 h \times 10^5$ N, and h is the broken height of the sand mudstone roof.

It can be seen that, when the sand mudstone roof is complete, $F > G_{coal}$. At the same time, the bearing capacity of the anchor cable is greater than the weight of the broken surrounding rock mass, and the roof remains stable. From $F > G_{coal} + G_{rock}$, when h > 2.02 m, the bearing capacity of the anchor cable is insufficient to break, the roadway roof collapses, and the ultimate failure depth of the hanging roof structure is 5.25 m.

As the working face advances forward, the top rock strata of the working face roadway will form an arched-shape hanging roof structure under the support of the section protection coal pillar and the end support, as shown in Figure 8.

The relationship between the horizontal dimension of the curved hanging roof structure and its thickness is [28]:

$$H = 1.96\gamma \cdot \alpha^2 / \sigma_S \tag{4}$$

In the formula, H is the thickness of the arched-shape hanging roof, and the anchor cable and the roof is taken as a whole, taking 6 m. $\alpha$ is the horizontal size of the arched-shape hanging roof, $\sigma_S$ is the unidirectional tensile strength of the coal body, taking 0.78 MPa, and $\gamma$ is the bulk density of the top coal arched-shape hanging roof, taking 13 kN/m$^3$.

The $\alpha$ can be obtained as 13.55 m.

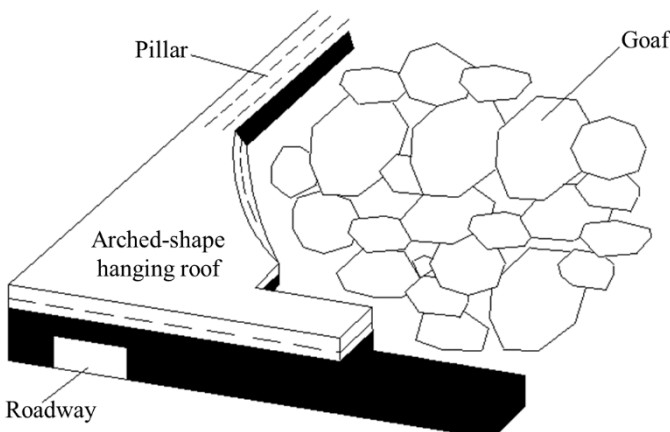

**Figure 8.** Arched-shape hanging roof at the end of mining roadway.

When the unidirectional tensile strength of the arched-shape hanging roof is constant, the greater the thickness of the arched-shape hanging roof, the greater the range of the arched-shape hanging roof. For the anchor-net support structure, as long as the anchor-net parameters are properly selected and the bolt (anchor cable) has sufficient preload, the low rock strata of the roadway roof are tightly connected by the "anchor-net-cable" structure to form a composite structure. At this time, the hanging roof structure can maintain a large strength, so it is easy to form an arched-shape hanging roof.

### 4.2. Numerical Simulation of Working Face End Hanging Structure

In order to analyze the influence of working-face excavation on the surrounding rock of a haulage roadway, especially the stress distribution of the surrounding rock of the haulage roadway and the characteristics of the plastic zone in front of and behind the working face under the action of mining stress, a FLAC3D numerical calculation model with a size of 500 m × 300 m × 80 m (length × width × height) is established according to the specific engineering geological conditions of the working face, and it is shown in Figure 9. We then apply displacement boundary conditions to the side and bottom of the model, and apply a vertical stress of 3 Mpa to the upper surface of the model.

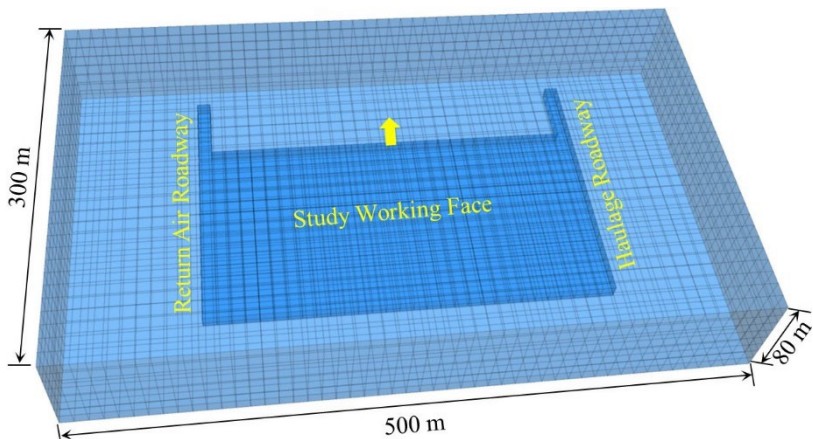

**Figure 9.** Numerical calculation model.

The numerical simulation of the roadway is full-section successive excavation, and the simulated excavation step is 1 m. The Mohr–Coulomb failure criterion has been adopted in this analysis. The physical and mechanical parameters of the model are shown in Table 2.

**Table 2.** Material parameters of each stratum in the model.

| Surrounding Rock | Thickness/m | Elastic Modulus /GPa | Poisson's Ratio | Cohesion /MPa | Internal Friction Angle/° | Dilation Angle/° | Density /kg/m³ | Tensile Strength /MPa |
|---|---|---|---|---|---|---|---|---|
| Overlying strata | 40 | 11.3 | 0.33 | 10.71 | 30.5 | 7.63 | 2550 | 2.8 |
| Mudstone | 4.5 | 11.5 | 0.17 | 8.60 | 32 | 8 | 2500 | 3.8 |
| Fine sandstone | 2 | 12.5 | 0.38 | 16.34 | 35.6 | 8.9 | 2950 | 7.52 |
| Sand mudstone | 4 | 12.18 | 0.18 | 9.15 | 33.49 | 8.37 | 2660 | 6.50 |
| No. 2 Coal | 6.5 | 2.37 | 0.26 | 2.21 | 34.14 | 8.54 | 1390 | 0.78 |
| Sand mudstone | 2 | 12.8 | 0.15 | 9.10 | 32.2 | 8.05 | 2650 | 4.18 |
| No. 3 Coal | 1 | 4.1 | 0.41 | 2.56 | 36.3 | 9.08 | 2750 | 1.38 |
| Mudstone | 1 | 11.5 | 0.17 | 8.60 | 32 | 8 | 2500 | 3.8 |
| Sand mudstone | 3 | 12.8 | 0.15 | 9.10 | 32.2 | 8.05 | 2650 | 4.18 |
| Fine sandstone | 5 | 12.5 | 0.38 | 16.34 | 35.6 | 8.9 | 2950 | 7.52 |
| Sand mudstone | 3 | 12.18 | 0.18 | 9.15 | 33.49 | 8.37 | 2660 | 6.50 |
| Fine sandstone | 8 | 12.5 | 0.38 | 16.34 | 35.6 | 8.9 | 2950 | 7.52 |

*4.3. Analysis of Numerical Simulation Results*

4.3.1. Plastic Zone and Stress Variation of the Roadway Surrounding the Rock in Front of the Working Face

The plastic zone and stress distribution of surrounding rock in front of the working face are shown in Figures 10 and 11.

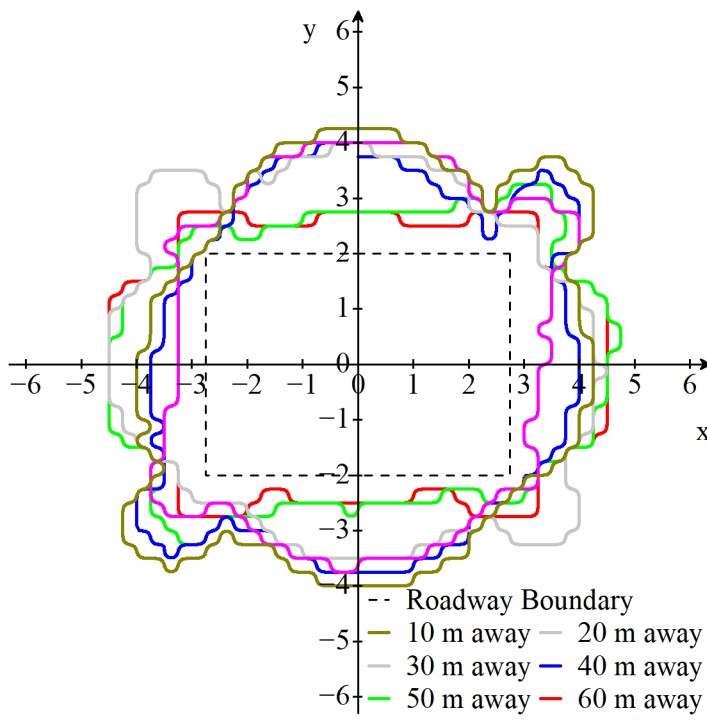

**Figure 10.** Distribution of the plastic zone of the surrounding rock of the roadway in front of the working face.

From Figure 10:

(1) As the distance from the working face gradually decreases, the range of the plastic zone of the roadway surrounding the rock gradually increases. When the range of the plastic zone around the rock mass reaches the maximum, the distance from the working face is 10 m and the maximum failure depth is 2.1 m. When the distance ahead of the working face is greater than 50 m, the range of the plastic zone of the roadway surrounding the rock tends to be stable, and the maximum failure depth is about 1 m.

(2) The failure process of the plastic zone of the roadway surrounding the rock shows obvious non-uniformity. When it is far away from the working face, the failure range of

the surrounding rock of the roadway side is larger than that of the roof and floor, and the failure is mainly concentrated in the center of the two sides. With the decrease of the distance from the working face, the rock layer at the shoulder of the roadway begins to be destroyed first, and the damage range and depth of the top gradually increase.

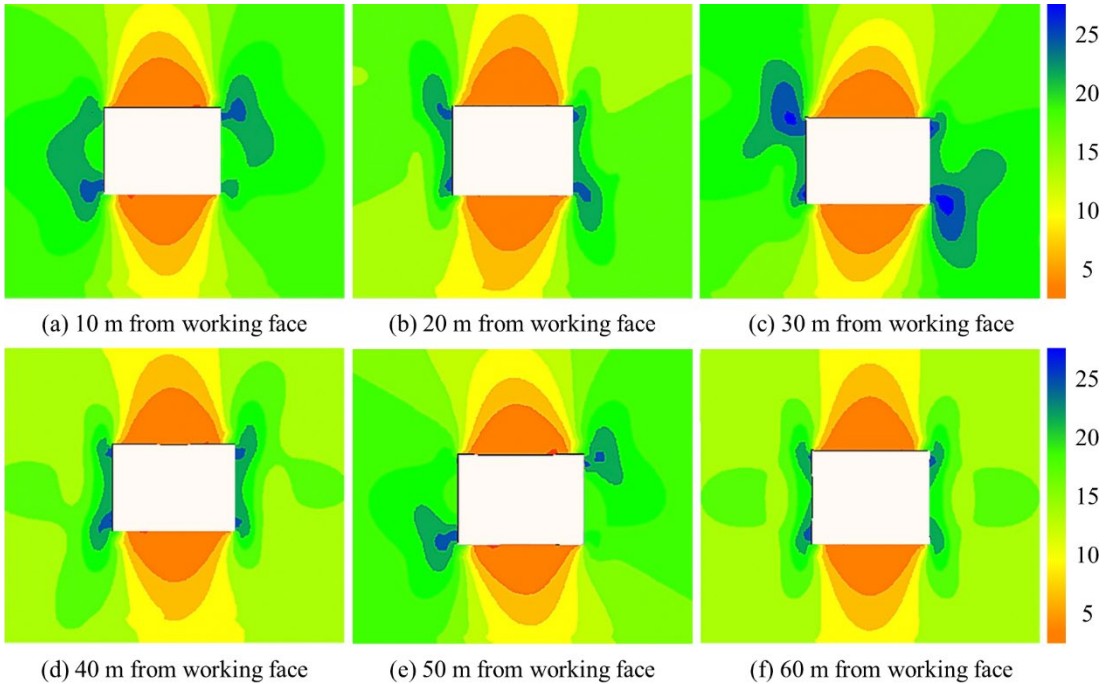

(a) 10 m from working face    (b) 20 m from working face    (c) 30 m from working face

(d) 40 m from working face    (e) 50 m from working face    (f) 60 m from working face

**Figure 11.** Vertical stress distribution of the surrounding rock of the roadway in front of the working face.

From Figure 11:

(1) Stress concentration occurs at the sharp corner of the roadway. As the distance from the working face decreases, the stress concentration phenomenon intensifies. The peak value of the stress concentration factor is 2.02, which is located near 10 m from the working face.

(2) The vertical stress of the roadway surrounding the rock shows obvious non-uniformity, and the stress concentration of the roadway sides and shoulders is more obvious than that of the roadway roof and floor. With the increase of the distance from the working face, the vertical stress of the roadway tends to be stable, and the stress concentration coefficient is stable at about 1.3.

In summary, in the range of the advancing working face, the plastic failure range of the roadway surrounding the rock gradually increases due to the influence of working-face mining. When the distance from the working face is 10 m, the failure range reaches the peak, and the maximum failure depth is 2.1 m, far from the ultimate failure depth of the hanging roof structure. Therefore, it is considered that, under the geological and supporting conditions, the disturbance of the working-face mining process has limited influence on the stability of the roadway.

### 4.3.2. Plastic Zone and Stress Change of the Roadway Surrounding the Rock behind the Working Face

The plastic zone and stress distribution of the surrounding rock of the roadway behind the working face are shown in Figures 12 and 13:

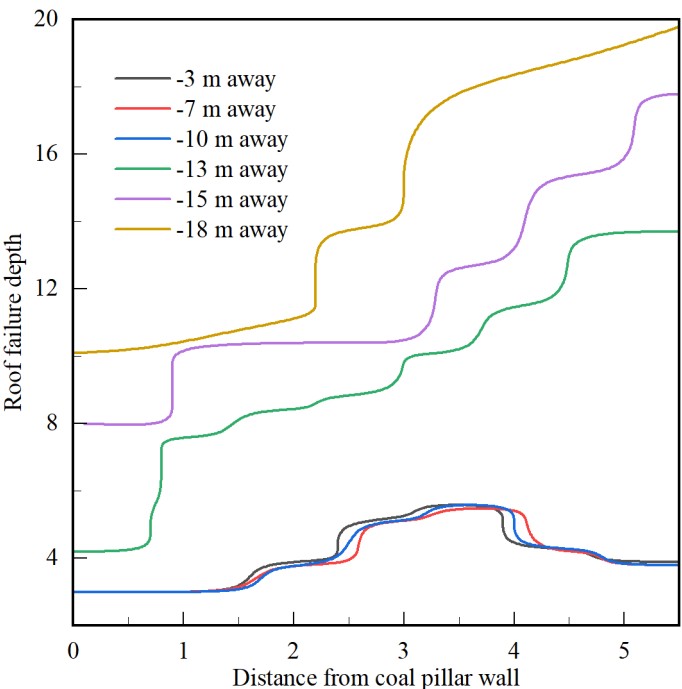

**Figure 12.** Depth of plastic zones around the roadway rock mass at various distances after the working face.

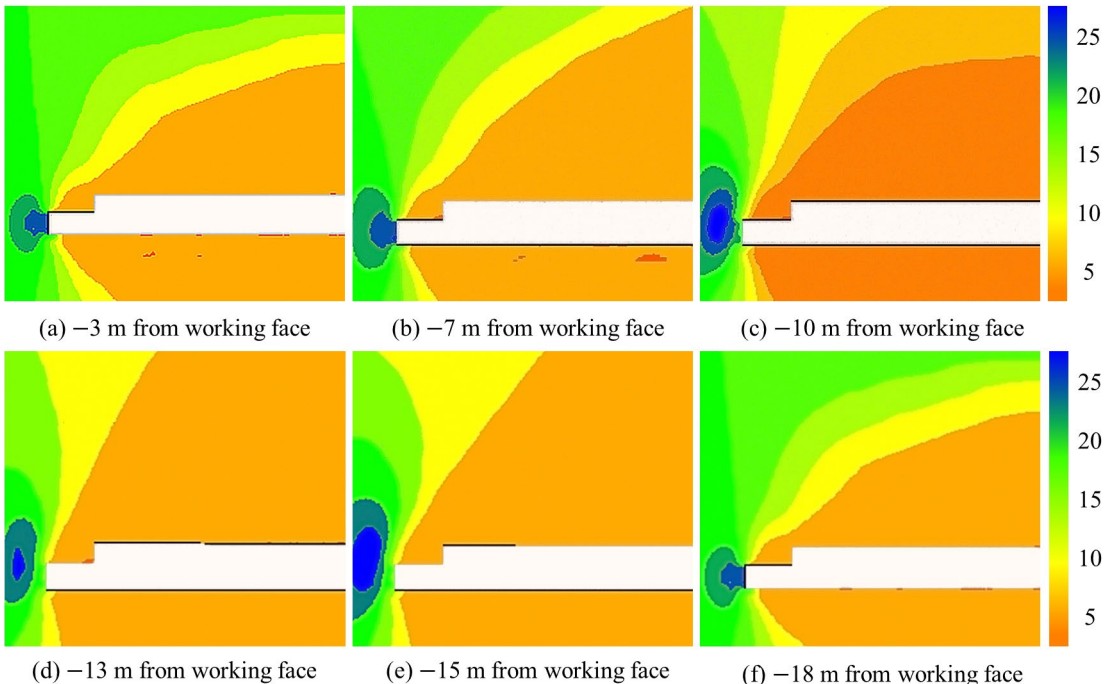

**Figure 13.** Vertical stress distribution of surrounding rock of the roadway behind the working face.

From Figure 12:

(1) As the working face advances forward, the failure depth of the rock strata at the top of the roadway increases slowly and the failure range changes near 13 m behind the working face. The failure depth increases from 2 m~3 m to 6 m~8 m. When the distance from the working face exceeds 15 m, the failure range of the roof in the goaf gradually stabilizes.

(2) In the range of 13 m behind the working face, it can be found that the maximum failure depth of the rock strata at the top of the roadway is 3 m, which does not reach

the calculated ultimate failure depth of the hanging roof structure. At the same time, the hanging roof distance of the roadway is 13 m, which is less than the caving length of the arched-shape hanging roof of 13.55 m, so the hanging roof structure is difficult to fall within this distance range.

(3) Outside the range of 13 m behind the working face, the failure depth of the roadway roof strata reaches more than 6 m, which has exceeded the ultimate failure depth of the hanging roof structure. At this time, the anchor cable breaks and the roof collapses. In addition, according to the calculation, when the span of the suspension structure exceeds 13.55 m, the suspension structure will break as a whole.

From Figure 13:

(1) The stress concentration phenomenon occurs in the protective coal pillar of the roadway behind the working face. As the working face advances forward, the stress concentration phenomenon intensifies, and the peak value of the stress concentration factor is 2.5, which is located 10 m behind the working face.

(2) When the distance from the working face exceeds 15 m, the vertical stress of the roadway surrounding the rock tends to be stable and the stress concentration factor is stable at about 1.8. At this time, the roof of the goaf behind the working face is completely collapsed, and the stress in the goaf is redistributed and gradually stabilized.

In summary, the numerical simulation results are consistent with the theoretical calculation results. In front of the working face, under the geological conditions and supporting strength, the rock surrounding the roadway remains stable. In the range of 13 m behind the working face, the hanging structure at the end of the working face remains stable. Outside the range of 13 m behind the working face, the anchor cable is not enough to bear the weight of the broken rock stratum of the roof, and the end hanging structure collapses and moves forward.

## 5. Discussion

*5.1. Caving Condition Analysis of Working Face End Hanging Roof Structure*

From the previous theoretical analysis, there are two forms of hanging roof structure collapse at the end of a coal mining face: one is that the roadway roof failure reaches the limit failure depth, the anchor cable breaks and fails, and the roadway roof falls; another case is that the length of the hanging roof structure of the roadway roof exceeds the limit span of the curved triangular plate. The numerical simulation results are analyzed, respectively.

(1) Failure depth analysis of roof

According to the theoretical analysis, when the destruction depth of the roadway roof exceeds 5.25 m, the roadway roof will fall under the weight of the rock mass in the broken area. Through numerical simulation analysis, the change of failure depth of the roadway surrounding the rock during mining can be obtained.

The maximum failure depth of the roadway roof strata in front of the mining face is 2.1 m. At this time, the broken area at the top of the roadway has not exceeded the anchoring range of the bolt. The roadway roof can still remain stable under the hanging of the bolt and anchor cable.

In the range of 13 m behind the working face, the damage depth of the roadway roof is less than 3 m. Outside the range of 13 m behind the working face, the damage depth of the roadway roof changes abruptly, reaching more than 6 m. At this time, the damage depth of the roadway roof has exceeded the anchoring range of the anchor cable, and the roof strata will fall due to the failure of the anchor cable. As the working face advances forward, the arched-shape hanging roof collapses periodically, and the new arched-shape structure gradually moves forward as the working face advances.

(2) Fracture analysis of arched-shape structure

According to the theoretical analysis, the two corners of the hanging arched-shape structure are on the two points behind the limit support or the coal pillar goaf. With the continuous advancement of the working face, the area of hanging roof structure increases, and the bending moment of hanging roof structure also increases. When the bending moment

reaches a specific limit, it breaks at the two positions of the coal pillar and the end, the end and the support roof cutting line, and the end hanging; in this process, the working face continues to move forward and a new arched-shape hanging roof is formed at the front end.

Through calculation, when the fixed edge of the hanging structure exceeds 13.55 m, the hanging structure will collapse. In the work site, the first damage occurs at the top of the hanging structure of these two places, and a hanging structure length of about more than 10 m will break the overall subsidence. After they are destroyed, a new limit will appear on the principal stress line, thereby forming an arched-shape limit bending moment trace.

Because the influence of the rock stratum above the anchor cable anchoring rock stratum is neglected in the theoretical calculation, the rock stratum of the anchor cable anchoring stratum will produce certain pressure when it collapses to the goaf, so the theoretical calculation distance is larger than the actual working face roadway. The existence of a certain range of arched-shape hanging roof is conducive to the stability of the roadway end and ensures the safety of the mining face. However, when the hanging roof area is too large, there will be potential safety hazards such as gas accumulation and large-scale roof collapse.

*5.2. The Mechanism of Working Face End Hanging Roof*

Based on the above research, the formation mechanism of the end hanging roof of the working face can be preliminarily obtained. The main conclusions are as follows:

(1) The lithology of sand mudstone and fine sandstone strata above the top coal of the roadway is strong. Under the influence of working-face mining, the mine pressure of the roadway is not severe. After the working face is advanced, the sand mudstone and fine sand strata above the top coal of the roadway are less damaged and the integrity is better.

(2) In the rear of the working face, the top coal and immediate roof above the roadway will be under the pressure caused by the overlying strata rotating to the goaf. Because the sand mudstone and fine sandstone above the top coal have strong lithology and strong supporting ability, the mudstone above the fine sandstone is not easy to form a long cantilever structure in the process of rotating to the working face; therefore, the sand mudstone and fine sand strata above the top coal are not easy to collapse under the support of the working face end support and the roadway coal pillar, forming a hanging roof structure. The length of the hanging roof structure depends on the mechanical properties of the top coal, sand mudstone, fine sandstone, and mudstone, and the stability of the structure formed by their combination.

(3) Through theoretical calculation and numerical simulation, the caving conditions of the hanging roof structure are analyzed. Combined with the actual situation of the working face, the caving step distance of the curved hanging roof structure is 10 ~ 13.55 m, and the curved hanging roof structure moves forward with the advance of the working face.

## 6. Conclusions

(1) After the mining roadway is affected by working-face mining, the main performance of two-side bulging is more obvious, roof and floor deformation is not obvious, the whole is less affected by mining, and strata behavior is not obvious. Through the observation of the roof strata of the roadway, it is found that the strata of the roof of the roadway do not change much, and the integrity of the whole rock strata is better. In the shallow part of the rock, fracture development or crushing zone appears, while the deep rock is relatively complete.

(2) There are two cases of collapse of the end suspension structure: The first case is that the roadway roof breaking depth is too large, anchor cable failure occurs, and hanging roof structure collapses; the second case is that the rock stratum of the roadway hanging roof structure breaks as a whole and collapses together with the goaf.

(3) Combining theoretical analysis and numerical simulation, the caving step of the arched-shape hanging roof structure is 10~13.55 m, and the hanging roof structure moves forward with the advance of the working face.

(4) After the above research, it is believed that the formation of the arched-shape hanging roof structure is due to joint influence of many aspects, as follows: the lithology of the roadway top coal is strong and the hanging roof structure is less affected by the mining of the working face; the rotary pressure of the mudstone above the fine sandstone is insufficient; the arched-shape hanging roof structure is formed under the joint work of the coal pillar and the end support.

Next, we will study the treatment method of the arched-shape hanging roof. It is expected to provide a method to control the size of the arched-shape hanging roof structure to keep it within a reasonable range, so as to ensure the safe production of the working face.

**Author Contributions:** H.L.: writing, methodology. C.H.: data curation, writing—original draft. Z.H.: investigation. Q.L.: numerical simulation. H.W.: data curation. J.L.: methodology. D.Z.: numerical simulation. All authors have read and agreed to the published version of the manuscript.

**Funding:** This research was funded by the National Natural Science Foundation of China (Grant No. 51574224), the National Natural Science Foundation of China (Grant no. 52004289), and the Fundamental Research Funds for the Central Universities (Grant no. 2022XJNY01).

**Institutional Review Board Statement:** Not applicable.

**Informed Consent Statement:** Informed consent was obtained from all subjects involved in the study.

**Data Availability Statement:** The data that support the findings of this study are available from the corresponding author, upon reasonable request. The authors gratefully acknowledge the help and comments of the reviewers and editors.

**Acknowledgments:** The authors gratefully acknowledge the test site provided by the China National Coal Group Corporation. The authors gratefully acknowledge the help and comments of the reviewers and editors.

**Conflicts of Interest:** The authors declare no conflict of interest.

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
