# Peer review of "Study on the Mechanism of a Hanging Roof at a Difficult Caving End in a Fully-Mechanized Top Coal Caving Face"

_sustainability, doi:10.3390/su15010812_

Round 1
Reviewer 1 Report
Reviewer’s introduction
The structural characteristics of the hanging roof at the end of the fully mechanised caving face have been determined through theoretical and numerical analyses in order to mitigate the risks (such as deformation, gas accumulation, and collapse) connected with the hanging roof of roadways behind the working face. Failure depth, break distance and formation mechanism of the hanging roof have been examined. The results show that mining activities has the minimal impact on hanging roof failure. Fine sandstone and mudstone lithologies covering the coal seams, however, were the main cause of failure. These lithologies' inadequate rotation pressure is the result of the joint effort of the coal support pillars. The break failure distance was between 10 and 14 m.
In my opinion, this study is interesting and should be considered for publication after the major revision.
Reviewer’s comments
1. Line 37: Verify that Kai [11] is the reference in place of Wang et al.
2. Line 83: What are the name and mining method of the coal mine in Shanxi Province?
3. Describe the fully mechanized cave mining method.
4. Line 95: citation for the geological column illustration in Fig. 1.
5. Line 97: give the definition of Platts coefficient? Justification Why was it taken? 1.5.
6. On top of the fine sandstone in Figure 2, the mudstone layer is not visible.
7. Label parts in Fig. 2.
8. Line 139: change the “the increase is increasing” to the deformation or convergence is increasing.
9. Line 140 should read, "sagging or subsidence of the roof and heaving of the floor," rather than "convergence of the roof and floor."
10. In Fig. 4:
- The roof and floor are what the legend depicts, not the roof and side walls.
- The Y-axis should be referred to as deformation or roof sagging or subsidence and floor heaving.
11. Line 156: the caption of Fig. 5 should be changed to “Detecting breaks and cracks at several monitoring stations placed apart from the working face and spaced by 20 m”.
12. Explain why the roadway roof displays a break and crack at 0.35 m and 2.70 m when it is 40 m from the working face but at 0.53 m and 4.1 m respectively when it is 20 m away.
13. Lines 165-167: “With the increase of the distance from the working 165 face, the integrity of the roadway roof is better, and no obvious broken zone and cracks”. Can you justify according to Fig. 5.
14. Lines 182 & 186 as well as Fig. 6: Legend, Sand mudstone not sandy mudstone.
15. Line 192: Where is the bolt's stress measured? Is this axial stress on the bolt head in Fig. 7?
16. What is the highest load or bolt bearing capacity that can be sustained before yielding?
17. In Fig. 7, Do these axial forces or loads that are carried by bolts at various distances from the working face exceed the bolt breaking limit or bolt capacity? Does this imply a reliable support system?
18. Y-axis, in Fig. 7, should be “Measured axial stress carried by bolt, ton”
19. Lines 203- 204, “With the increase of the distance from the working face, the stress of the bolt tends to be stable at about 2 t.” . Please check, because bolt No. 3 carries stress of 3 ton at 40 m from the working face.” The sentence might be changed to read: “induced stress on the bolt falls as distance from the working face increases”.
20. Lines 205-206, “the stress of the bolt (e.g., No. 4) in the middle of the of the roadway roof is greater than that on both sides (e.g., No. 2 and No. 5) of the roadway roof”. See Fig. 2.
21. Lines 208-209, “the stress of some bolts (e.g., No. 2, No. 3 and No. 5) on the roof of the roadway tends to be stable at about 3 t”.
22. Lines 264-269, delete apostrophe “,” at the end of Equations (1-4).
23. Line 272: g is the gravity acceleration value, take 9.8.
24. Lines 275-276: G coal = 1390 × 5.6 × 3.23 × 1× 9.8 N = 2.46× 105 N,
25. In Equation 6, why are there four dimensions for the volume? Grock = 2660 × 5.6 × 3.23× 1× h× 9.8 N = 1.46× 105 hN,
26. If h is unknown, Equation 6 should be written as: Grock = 1.46 h× 105 N
27. Line 283: In the entire manuscript, the phrase "arc-shaped hanging roof" should be replaced with "arched-shape hanging roof."
28. Remove "apostrophes" from all of the manuscript's equations.
29. Line 290: Do you mean “α” instead of “A”? “A is the horizontal size of the arc hanging roof structure”.
30. σ S should be retyped as: бS
31. Line 293:
- Rewrite the equation using Equation Word Editor as follows:
-
- Check your numbers, according to your assumption that, the thickness of arched-hanging roof structure is 6 m not 7 m as per equation 8.
32. Line 309: Which software was used to build the numerical model? Is it FLAC?
33. What are the model boundary conditions (BCs) for the model?
34. Did you apply fine-mesh size for the working face area to catch stresses?
35. Line 315: Cite the geo-mechanical model input parameters listed in Table 2.
36. It would be preferable to provide a plan-view next to Figure 9 that shows the various strata that have been modelled.
37. Line 313: “The model is Mohr-Coulomb model”. This sentence would be preferrable to alter to read as: Mohr-Coulomb failure criterion has been adopted in this analysis.
38. When using Mohr-Coulomb yielding function, the dilation angle (ψ) should be used. ψ = φ/4 (where φ is the internal friction angle). In Table 2, add another column to show the dilation angles for each stratum.
39. Why you choose plastic zones as failure evaluation criterion? Why did you not choose deformation to estimate roof sag and floor heave?
40. Did you determine the length (extent) of plastic or yield zones against the anchorage length of rock support?
41. What is the threshold (limit) of yielding zone as failure condition?
42. It would be clearer and more interesting if a single 2D line graph has been inserted to summarize the results of evolution (development) of the depth of yielding zones around roof, floor, right and left sidewalls at different distances (e.g., 10, 20, 30, 40, 50 and 60 m) from the working face.
43. The legends for the vertical stress distribution in Fig. 11 are unclear. It is advised to either incorporate a single 2D bar chart to illustrate the development of vertical stresses around the roof, floor, and sidewalls at various distances from the working face, or to replace the existing figure with one that is clearer.
44. What do you mean by stress concentration factor “SCF” in Line 341? In terms of stability, is there a SCF threshold?
45. The caption of Fig. 12 is suggested to be written ass follows: “Depth of plastic zones around roadway rock mass at various distances from the working face”.
46. It would be clear if a 2D line graph depicts the extent of yielding zones around roadway rock mass at different distances from the working face.
47. Line 391: “the roadway surrounding rock”- Do you mean “the rock surrounding roadway”?
48. Lines 409-410: What is the anchorage of the rock bolt ?

Reviewer 3 Report
The research content of this paper is relatively rich, but the research significance is not clear. The summary of previous achievements is not clear enough and lacks logic. It does not meet the employment standards of the journal.
Author Response
Dear Reviewer 3:
Thank you for your patience concerning our manuscript entitled “Study on the Mechanism of Hanging Roof at Difficult Caving End in Fully Mechanized Top Coal Caving Face”. (ID: 2050075).
Thank you for your time and patience in reviewing my manuscript. We have revised the paper and look forward to your comments.
Reviewer 4 Report
This manuscript studies the mechanism of arc triangle roof at the end of mining roadway in the process of fully mechanized top coal caving face mining, and introduces the potential safety hazards that arc triangle roof is easy to cause. Based on a case, the mechanism is studied by using field investigation, theoretical analysis, and numerical simulation methods. This paper expounds the mechanism of arc triangle roof, which provides a theoretical basis for roadway roof control in the process of fully mechanized top coal caving, and has important guiding significance. This manuscript is detailed, accurate and innovative, and it should be ready for publication with minor revision in my opinion.
My comments are as follows:
(1) In the abstract, “It is concluded that the hanging roof structure is due to the strong lithology of sand mudstone and fine sandstone above the top coal of the roadway, the hanging roof structure is less affected by the mining of the working face, and the rotation pressure of the upper mudstone is insufficient, and it is formed by the joint work of the protective coal pillar and the end support.”, the sentence is too long to be understood.
(2) In the introduction, the last paragraph of the introduction is not clearly explained the aim of the paper. Please re-write it.
(3) In the Table 1, the “sand mudstone” is different from that in other parts such as Figure 6 of the manuscript, which is called “sandy mudstone”. Please check the full text.
(4) The form of the formula such as formula (8) in this paper is not standard, please correct it.
(5) In the section 5.1, “The existence of a certain range of arc triangular plate structure is conducive to the stability of the roadway end and ensures the safety of the mining face. However, when the hanging roof area is too large, there will be potential safety hazards such as gas accumulation and large-scale roof collapse.” Should this sentence appear in the introduction instead of here.
(6) Some of the sentences in this manuscript have language problems. Please check the full text and modify them.

Round 2
Reviewer 1 Report
Thanks authors for addressing all comments. I have no further to add.
Reviewer 2 Report
Dear authors the paper is ready for publication. Good work!
Reviewer 3 Report
Accept in present form